# Does the Environmental Regulation Intensity and ESG Performance Have a Substitution Effect on the Impact of Enterprise Green Innovation: Evidence from China

**DOI:** 10.3390/ijerph19148558

**Published:** 2022-07-13

**Authors:** Fengyan Wang, Ziyuan Sun

**Affiliations:** 1School of Public Policy & Management, China University of Mining and Technology, Xuzhou 221116, China; 31012@sdwu.edu.cn; 2School of Accounting, Shandong Women’ University, Jinan 250300, China; 3School of Economic and Management, China University of Mining and Technology, Xuzhou 221116, China

**Keywords:** environmental regulation intensity, green innovation, ESG performance

## Abstract

Against the background of green and sustainable development strategy, it is an effective way to carry out green innovation to cope with the increasing intensity of government environmental regulation for enterprises. Nevertheless, the regulatory role of ESG performance has been ignored. Based on panel data from Chinese listed companies from 2010 to 2019, this paper mainly studies whether the environmental regulation intensity and ESG performance have a substitution effect on the impact of green innovation by constructing a double fixed effect model. The empirical results showed that first, positive ESG performance is conducive to promoting green innovation. Second, there is a U-shaped relationship between the intensity of environmental regulation and high-quality green innovation, which reflects the effect of “offset before compensation”. With the increasing intensity of environmental regulation, high-quality green innovation tends to crowd out low-quality green innovation, which further improves the practical test of the “Porter Hypothesis”. Third, the positive ESG performance showed a negative regulatory effect between environmental regulation intensity and enterprise green innovation, which means that environmental regulation intensity and ESG performance have a substitution effect, and the effect is heterogeneous in different enterprises. This paper makes a beneficial exploration on how environmental regulation intensity and ESG performance affect enterprise green innovation, and demonstrates the regulatory role of ESG performance between environmental regulation intensity and green innovation, which reveals the impact of macro environmental policies on the green innovation behavior of micro subjects, and contributes to the further improvement of ESG concept and green innovation theory.

## 1. Introduction

Since humanity entered the 21st Century, climate change, environmental pollution, and other issues affecting sustainable economic and social development have become increasingly prominent. The Chinese government has successively introduced a number of environment-related systems and measures around its green and sustainable development strategies in recent years, such as administrative measures for the legal disclosure of enterprise environmental information, administrative measures for the transfer of hazardous waste, and the decision to abolish the relevant regulations and normative documents on the import of solid waste. These environmental regulations contain stricter requirements for enterprise environmental emissions, and the intensity of environmental regulation is gradually increasing [1]. In this context, enterprises are facing increasingly stringent environmental emission control systems, which will inevitably bring higher environmental emission costs to enterprises. From a strategic point of view, the increase in the intensity of the government’s environmental regulation will exacerbate the scarcity of resources in the capital market [2], which will urge enterprises to invest limited funds in high-quality green innovation to quickly improve productivity and profitability [3], so as to cope with the increasing cost of environmental regulation, thus helping to promote green innovation of enterprises.

The Environmental Social and Governance (ESG) concept proposes requirements for the strategic objectives of enterprises from three aspects: environmental responsibility, social responsibility, and corporate governance, which has become an important indicator generally recognized by all countries to measure the sustainable development of enterprises under the green development model [4,5]. In order to reduce environmental costs from a long-term perspective and achieve green and sustainable development, many enterprises increase the trust of institutional investors by actively disclosing ESG information so as to alleviate their financing constraints [6]. For institutional investors, increasing R&D investment in enterprises with excellent ESG performance can not only ensure the safety of investment funds, but also benefit from the sustainable development of enterprises [7]. ESG provides a set of operable and practical sustainable development evaluation systems [8], which helps to optimize the market investment structure [9], so as to guide the flow of social capital to ecological, green, and low-carbon fields, and help to stimulate the green innovation behavior of enterprises. Thus, it is worth noting whether the increasing intensity of environmental regulation has promoted high-quality green innovation in enterprises? Does ESG performance have a positive impact on green innovation? And does ESG play a regulatory effect on the intensity of environmental regulation and green innovation? What are the differences of this effect in different types of enterprises? These are the key problems to be solved in this paper.

The main contributions of this paper are three: (1) this paper makes a beneficial exploration on how ESG performance affects enterprise green innovation and its regulatory role between environmental regulation intensity and green innovation, which broadens the scope of enterprise ESG research and contributes to the further improvement of ESG concept and green innovation theory. (2) This paper explores the heterogeneous impact of environmental regulation intensity on different quality green innovations of enterprises. It is found that with the continuous increase of environmental regulation intensity, high-quality green innovation has a crowding-out effect on low-quality green innovation, making low-quality green innovation decline, thus promoting high-quality green innovation, which further improve the practical test of “Porter Hypothesis”; (3) Through further testing by industry and region, this paper analyzes the heterogeneous impact of environmental regulation intensity on green Innovation of different types of enterprises, which provides a reference for the government to make targeted environmental regulation decisions.

The main structure of this paper is as follows: the second chapter states the research hypotheses, the third chapter explains the research design and model construction, the fourth chapter describes an empirical test of the relationship between environmental regulation, ESG performance, and green innovation, and the fifth chapter summarizes the full text and proposes ways to optimize ESG performance suggestions to formulate precise environmental regulations to improve green innovation in enterprises.

## 2. Literature Review

### 2.1. ESG and Green Innovation

ESG is a new concept emerging in the field of sustainable development in recent years. In the research of ESG, the existing literature mainly focuses on analyzing the economic consequences of ESG information disclosure [9,10] and ESG investment [11,12], with few literature papers considering the regulatory effect between environmental regulation and green innovation. There are few existing research results to study its impact on green innovation from the overall level of ESG, mainly from the perspectives of environmental responsibility, social responsibility, and corporate governance. For example, from the perspective of environmental responsibility, enterprises’ active commitment to environmental social responsibility will significantly increase investment in green innovation, and the government’s environmental regulatory pressure will promote enterprises’ green innovation behavior [13]. Therefore, the government should encourage enterprises to assume more environmental responsibility [14]. By actively disclosing the information about their environmental responsibility, enterprises can enhance the public’s recognition of the enterprise’s environmental awareness, and then promote the enterprise’s green innovation [15]. From the perspective of social responsibility, enterprises actively fulfill their social responsibility, which can effectively improve the sustainable development of the environment, and then promote green innovation [16,17]. Better social responsibility can also help enterprises obtain funds to improve production technology and service quality, so as to improve their market competitiveness and increase shareholders’ wealth [18]. From the perspective of corporate governance, corporate governance factors such as a good management system [19] and internal control level [20] significantly encourage green innovation.

### 2.2. Environmental Regulation Intensity and Green Innovation

Researchers have conducted studies on the impact of environmental regulation intensity on enterprise green innovation from different perspectives, but they have not reached an agreement. The first view is that environmental regulation has a linear impact on green innovation, mainly positive or negative. Most researchers believe that environmental regulation has a positive impact on green innovation. For example, Lanoie et al. took OECD countries as research samples and found that environmental regulation does have an incentive effect on innovation [21]. Kneller and Manderson took the UK manufacturing industry as an example and found that environmental regulation promoted green innovation, but had no significant impact on non-green innovation [22]. Rubashkina et al. believe that the strengthening of environmental regulation of the European manufacturing industry has increased the number of green patents and the level of innovation [23]. Quan et al. also believe that public participation and formal environmental regulations have promoted industrial technological innovation [24]. Jiang et al. has reached a similar conclusion. It is believed that regional environmental regulation can promote a high level of innovation in Chinese enterprises [25]. Some studies also state that environmental regulation is not conducive to enterprise green innovation. For example, Gray believes that mandatory environmental regulation has had a negative impact on green innovation in the American manufacturing industry [26]. Dean et al. performed an econometric analysis of Swedish manufacturing data from the cost perspective and found that strict environmental regulation hindered enterprises from carrying out green innovation [27].

Some researchers believe that the impact of environmental regulation on enterprise green innovation will change with time and the environment. Lanoie took the Canadian manufacturing industry as an example and found that the long-term dynamic impact of environmental regulation on productivity was positive, but the short-term impact was negative [21]. According to Zhu et al., there is a relationship between market-incentive environmental regulation and green innovation in high-tech enterprises [28]. Some also hold the view that environmental regulation does not affect green innovation. For example, Nakano showed that environmental regulation did not significantly affect technological innovation in the Japanese paper industry [29]. The empirical results of He et al. also showed that environmental regulation does not directly affect green innovation performance [30]. In addition, many scholars have studied the role of different types of environmental regulation. For example, environmental regulation is divided into command and control [31,32], market-based [33,34], Mandatory [35,36], and formal and informal [25,37] etc.

This existing research provides the basis for this article, but some deficiencies remain to be addressed. First, the existing research mainly focuses on the impact of single variables of environment, society, and governance on enterprise innovation, and few literatures consider the impact of ESG comprehensive performance on green innovation. Second, existing studies mainly focus on the impact of different countries and types of environmental regulations on green innovation, and differentiae the types of environmental regulation, and are less involved in heterogeneity analysis of the quality of green innovation. Third, less attention has been paid to the relationship among ESG performance, environmental regulation, and green innovation. Based on the deficiencies of the above research, this paper uses the data of Chinese listed companies from 2010 to 2019 to explore whether the environmental regulation intensity and ESG performance have a substitution effect on the impact of green innovation by constructing a fixed effect model, which strives to provide new evidence for the further expansion and application of Porter Hypothesis.

## 3. Research Hypotheses

### 3.1. Environmental Regulation Intensity and Enterprise Green Innovation

When an enterprise operates in an area with low environmental regulation intensity, where environmental emission regulations are relatively loose, the cost of being environmentally responsible is low. Under low-intensity environmental regulation, enterprises face less environmental compliance costs, and reducing environmental emissions is more in line with the principle of cost-effectiveness, which makes enterprises increase the investment in purchasing environmental protection equipment and reduce the investment in technological innovation, which is not conducive to improving the driving force of technological innovation of enterprises [26,27]. When the government’s environmental regulation intensity gradually increases and becomes sustainable, enterprises will face the pressure of gradually increasing environmental costs. Relying on the purchase of environmental protection equipment to reduce environmental emissions and other measures will be a difficult way to offset the increasing cost of environmental regulation. At this time, the implementation of green innovation by enterprises to improve production efficiency and technical sophistication will be more in line with their long-term interests. Green innovation can reduce the environmental protection cost of enterprises, improve production efficiency, and increase competitiveness from a long-term perspective [23,38], to stimulate the green innovation of enterprises. Therefore, it is assumed that:

 **Hypothesis 1a (H1a).**
*The impact of environmental regulation intensity on green innovation presents a U-shaped relationship, which reflects the effect of “offset before compensation”.*


In previous research and practice, green innovation can be divided into high-quality and low-quality. Generally speaking, high-quality green innovation (such as green patent technology) can effectively improve the technical sophistication and production efficiency of enterprises, and its performance in realizing sustainable enterprise development is better than that of low-quality technological innovation (such as a utility model) [39]. Considering the differences between high-quality and low-quality green technology innovation in improving production efficiency and profitability, due to the scarcity of research and development funds, if the intensity of environmental regulation continues to increase, enterprises will focus research and development funds on high-quality green innovation and offset high environmental cost pressures by improving profitability [38]. Therefore, the following assumption is made:

 **Hypothesis 1b (H1b).**
*There is a U-shaped relationship between the intensity of environmental regulation and high-quality green innovation.*


In fair market competition, innovation resources are always scarce [40], and enterprises’ green research and development (R&D) investment is bound to be limited. From the perspective of cost and benefit, when the environmental regulation intensity of the government is weak, the environmental compliance cost of enterprises is low. Compared with green innovation activities that require a lot of R&D funds, enterprises will choose to reduce terminal emissions at a lower cost [23]. Therefore, they have a negative attitude towards high-quality and low-quality Green Innovation. However, when the intensity of environmental supervision exceeds the critical value, from a long-term perspective, enterprises tend to invest in high-quality green innovation to offset environmental costs, further limiting the investment in low-quality Green Innovation (such as utility models), thus reducing the output of low-quality green innovation [37]. Therefore, the following assumption is made:

 **Hypothesis 1c (H1c).**
*With the increasing intensity of environmental regulation, low-quality green innovation is decreasing.*


### 3.2. ESG Performance and Enterprise Green Innovation

ESG performance can be divided into environmental, social, and governance performance. First, from the perspective of environmental performance, in order to realize the sustainable development of resources and the environment, enterprises should pay more attention to the demands of the government, the public and other stakeholders for green sustainable development while meeting the interests of shareholders. The enterprises are also supposed to reduce environmental costs as much as possible and strengthen the expression of environmental responsibility. In order to achieve the goal of sustainable development, the improvement of the level and quality of green innovation is an inevitable choice for enterprises [41]. Second, from the perspective of social responsibility performance, the demands, expectations and objectives of the stakeholders of the enterprises are not consistent [42,43]. Good social responsibility performance can convey the signal of actively serving the society to the government, investors and the public, so as to effectively alleviate the regulatory pressure of the government and relevant departments on the enterprises. As well, enterprises can obtain special policy support, tax incentives and financial capital incentives from the government to engage in green innovation, and gain access to diversified innovation knowledge resources which helps to improve the green innovation ability of enterprises [44,45]. Third, from the perspective of governance performance, good corporate governance is the performance of enterprise management norms which belongs to the internal resources of the enterprise. The external resources of stakeholders can also be combined with the internal resources of the enterprise to realize resource exchange and mutual assistance, and provide policy and technical support for enterprise green innovation [19], which contributes to improving enterprise green innovation. Therefore, the following assumption is proposed:

 **Hypothesis 2 (H2).**
*When other conditions are certain, positive ESG performance is conducive to promoting green innovation.*


### 3.3. Regulatory Effect of ESG Performance

The ESG concept advocates the unity of corporate social value, environmental responsibility, and economic value. The better the ESG performance of the enterprise, the stronger will be its willingness and ability to fulfill environmental and social responsibilities, and the higher will be its level of corporate governance. In addition, the enterprise is expected to have greater flexibility to deal with environmental regulation under turbulent market conditions [6,46], which shows that the enterprise itself has a certain level of innovation capability and the ability to fulfill its responsibilities. It can also send a signal to the government to actively fulfill its responsibilities to creditors, employees, the public, and other stakeholders to raise the expectations of government environmental regulation for ESG. In practice, companies with better ESG performance pay more attention to the interests of shareholders, employees, the public and other stakeholders and the company’s social image, and their input and output level in green innovation is high. Therefore, with the increase of the degree of environmental regulation, these enterprises are less likely to have environmental compliance problems, so they can better resist the pressure of environmental expenditure cost caused by the increase of the intensity of environmental regulation. In other words, environmental regulation intensity and ESG performance have substitution effects in promoting green innovation. Therefore, the following assumption is proposed:

 **Hypothesis 3 (H3).**
*ESG performance has a negative regulatory effect on the impact of environmental regulation and green innovation, which means that environmental regulation intensity and ESG performance have substitution effect in promoting green innovation.*


Based on the above analysis, the theoretical model of this paper is shown in Figure 1.

## 4. Sample Sources and Empirical Models

### 4.1. Sample Sources

Based on the availability and accuracy of data, this study used the data for Listed Companies of China from 2010 to 2019 as the research sample. Among them, the enterprise green innovation data were obtained by matching the research data for green patents of Listed Companies in the CNRDS database with the data for green invention patents and utility model patents published by the State Intellectual Property Office. The ESG data came from Bloomberg’s environmental, social, and governance database; environmental regulation data came from the China Environmental Statistics Yearbook and the China Statistical Yearbook, and the missing data were obtained by consulting the statistical yearbook and the government website of each province, and data for other control variables came from the CSMAR database and the wind database. To ensure the reliability of the sample data, this study excluded financial industry, enterprises with abnormal operation status, and the enterprises for which samples could not be obtained for the selected variables. Winsorizing was then carried out on the 1% and 99% quantiles of the main variables.

### 4.2. Variable Definition

#### 4.2.1. Independent Variable

Referring to the practice of Zhao et al. [47] and others, this paper uses the regional environmental regulation intensity index to measure the intensity of environmental regulation (ERI). The emissions of industrial wastewater, industrial SO_2_, and industrial smoke and dust from all provinces (municipalities and autonomous regions) were selected, and the entropy method was used to calculate the regional environmental regulation intensity index. The environmental regulation intensity index was calculated as follows:

First, the extreme value method was used to standardize the pollutant discharge in different regions to enhance data comparability:$$P_{i,j}=\frac{M_{i,j}-Min\left(M_{i,j}\right)}{Max\left(M_{i,j}\right)-Min\left(M_{i,j}\right)}$$
where *M_i,j_* is the initial value of the *j*-th pollutant emission in the *i*-th region and max (*M_i,j_*), min(*M_i,j_*) are the maximum and minimum of the *j*-th pollutant in the current year in the *i*-th region respectively.

Second, the adjustment coefficient is calculated by regions according to the pollutant category:$$W_{i,j}=\frac{M_{i,j}}{Mean\left(M_{i,j}\right)}$$
where Mean (*M_i,j_*) is the average of the emissions of the *j*-th pollutant in region *i*. The pollutant adjustment factor is used to distinguish the degree of environmental governance in different regions.

Third, the overall environmental emissions intensity (*Q_i_*) is calculated by region.
$$Q_{i}=\frac{1}{3\sum P_{i,j}\times W_{i,j}}$$

Fourth, the overall environmental emission intensity is positively treated to obtain the regional environmental regulation intensity (ERI):$$ERI_{i}=\frac{Max\left(Q_{i}\right)-Q_{i}}{Max\left(Q_{i}\right)-Min\left(Q_{i}\right)}$$

#### 4.2.2. Dependent Variable

Referring to the practice of Zhu et al. [28] and others, this study selects the number of green patent authorizations to measure green innovation (GIA) because green patent applications are not necessarily authorized. According to the definition of China’s Patent Law, invention patents focus on the technological innovation of products and methods, whereas utility models prefer the practical application of product shape and structure, which shows that invention patents are more effective in improving the level of technological innovation. Therefore, this article uses green invention patents to measure high-quality green innovation (HGIA), and low-quality green innovation (LGIA) is measured by green utility model patents.

#### 4.2.3. Moderator Variables

The ESG performance is an important standard for the international community to measure the level of green and sustainable development of enterprises. As the world’s leading business information provider, Bloomberg offers corporate ESG data including ESG comprehensive scores, environmental scores, social responsibility scores and governance scores, which has a certain authority in the industry. As an important indicator to measure the sustainable development ability of enterprises, ESG index comprehensively reflects the performance of enterprises in protecting the environment, fulfilling social responsibility and corporate governance. Therefore, based on the practice of Zhang et al. [10], this paper uses the ESG index developed by Bloomberg to measure the ESG performance of enterprises. Specific indicators are shown in Table 1.

#### 4.2.4. Control Variables

Referring to the research of Quan et al. [24], Zhu et al. [28] and Jiang et al. [25], financial leverage (LEV), profitability (ROA), corporate value (TQ), growth ability (GRO), two duty unification (DUA), proportion of independent directors (PID), composite tax rate (CTR) and property rights (PRO) are selected as control variables.

Financial leverage: Generally speaking, the asset-liability ratio represents the solvency of an enterprise which is generally used to measure the financial leverage of enterprises. If an enterprise has insufficient funds, it will affect the amount of funds invested in green innovation activities, thus affecting the output of green innovation.

Profitability: When the profitability is high, enterprises will have more funds to invest in green R&D activities, which will have a positive impact on green innovation.

Corporate value: Corporate value represents the accumulated wealth and future value of the business entity. Generally speaking, enterprises with high enterprise value will pay more attention to the planning of future strategies and the sustainability of business activities, so they will pay more attention to innovation activities.

Growth ability: Growth capability represents the ability and potential of an enterprise to continuously make profits to achieve quantitative expansion and qualitative improvement. Enterprises with strong growth capability usually pay more attention to R&D activities and social responsibility, and their performance in green innovation will also increase. In practice, operating profit margin or operating income growth is considered to be an important factor affecting enterprise innovation. Yang et al. (2010) [48] and Jiang et al. (2018) [8] and other researchers have confirmed this conclusion. From this point of view, the control index of growth ability has been set to eliminate this problem.

Two duty unification: If the chairman and CEO are held by the same person, if the chairman and CEO are held by the same person, the corporate governance structure will be weakened, and the enterprise decision-making will lack democracy, which will have an adverse impact on green R&D.

Proportion of independent directors: When the proportion of independent directors is high, it indicates that the board of directors has strong independence, which can promote the improvement of enterprise performance and have a positive impact on green innovation.

Composite tax rate: When the comprehensive tax burden of an enterprise is low, it means that the enterprise enjoys more tax preferences, which will save more funds for green innovation activities, thus having a positive impact on green innovation.

Property rights: It is generally believed that the nature of property rights, such as whether it is a state-owned enterprise, will affect the strategic decision-making and operation management of enterprises, thus having a heterogeneous impact on green innovation.

Based on the above considerations, this paper takes the above as control variables to eliminate their impact on the results.

The main variable design is shown in Table 2.

### 4.3. Model Construction

To test the impact of environmental regulation on high-quality green innovation and overall green innovation in enterprises, this study constructed the following model:(1)$$GIA_{i,t}=\alpha_{0}+\alpha_{1}ERI_{i,t}+\alpha_{2}ERI_{i,t}{}^{2}+\alpha_{3}Controls_{i,t}+\theta_{i}+\delta_{i}+\varepsilon_{i,t}$$
where *Controls_i,t_* represents all control variables.

To test the impact of environmental regulation on low-quality green innovation in enterprises, this study constructed the following model:(2)$$GIA_{i,t}=\beta_{0}+\beta_{1}ERI_{i,t}+\beta_{2}Controls_{i,t}+\theta_{i}+\delta_{i}+\varepsilon_{i,t}$$

To examine the impact of ESG comprehensive performance, environmental responsibility performance, social responsibility performance, and corporate governance performance on green innovation, this study constructed the following model:(3)$$GIA_{i,t}=\mu_{0}+\mu_{1}ESG_{i,t}+\mu_{2}Controls_{i,t}+\theta_{i}+\delta_{i}+\varepsilon_{i,t}$$
where *ESG_i,t_* represents comprehensive ESG, including ERP, SRP, and GP.

To test the moderating effects of ESG on overall performance, environmental responsibility, social responsibility, and corporate governance in the impact of environmental regulation on corporate green innovation, this study constructed the following model:(4)$$GIA_{i,t}=\gamma_{0}+\gamma_{1}ERI_{i,t}+\gamma_{2}ERI_{i,t}{}^{2}+\gamma_{3}ERI_{i,t}ESG_{i,t}+\gamma_{4}ERI_{i,t}{}^{2}ESG_{i,t}+\gamma_{5}Controls_{i,t}+\theta_{i}+\delta_{i}+\varepsilon_{i,t}$$

In model (1)–(4), *α_0_*, *β_0_*, *γ_0_*, *μ_0_* are the intercept, *α*_1*~n*_, *β*_1*~n*_, *γ*_1*~n*_, *μ*_1*~n*_ are the coefficient, *θ_i_* is the fixed effect of the industry to which the enterprise belongs, *δ_i_* is the individual fixed effect of the enterprise, and *ε_i,t_* is the residual term.

## 5. Empirical Results

### 5.1. Descriptive Statistics

The descriptive statistics of the main variables were completed by Stata 16.0 (Stata Corp., College Station, TX, USA). Table 3 presents the descriptive statistics for the main variables. The average GIA of the sample companies was 0.2984, the maximum value was 6.7708, and the minimum value was 0, indicating that there is a large difference in the green innovation level of listed companies in China; the average value of ERI was 0.7197, indicating that different provinces show wide variations in the level of environmental regulation (cities and districts) and that the sample companies are more distributed in areas with strong environmental regulations. The mean values of ERP, SRP, and GP were 2.2869, 3.1110, and 3.8178, and the standard deviations were 0.5875, 0.4039, and 0.1115, respectively, indicating that there are large variations in ERP and SRP among enterprises, but only small differences in GP, mainly because the CSRC has mandatory requirements for corporate governance. The variance inflation factors (VIF) of the variables were all less than 10, and the average value was 1.26, indicating that there was no high degree of collinearity among the variables.

The kernel density chart of ESG performance of sample enterprises is drawn by using stata16. It can be seen from Figure 2 (on the left) that the peak of the nuclear density curve changes from steep to gentle over time, indicating that the difference in ESG performance among enterprises gradually increases. In addition, the wave crest of the ESG curve gradually moves to the right, which indicates that the score of ESG performance increases, and the enterprise’s ESG performance shows a growing trend. From Figure 2 (on the right), the nuclear density curve of environmental regulation intensity shows a multi peak distribution, indicating that there is a phenomenon of multipolar differentiation in environmental regulation intensity among regions.

### 5.2. Empirical Results

#### 5.2.1. Analysis of the Impact of Environmental Regulation on Enterprise Green Innovation

Table 4 (1) shows that the influence coefficients of the square of environmental regulation intensity and of environmental regulation intensity on green innovation were 0.268 and −0.394 respectively, which were significant at the 1% and 5% levels, respectively, and their relationship has passed the U test (In order to test whether the U-shaped relationship between variables is valid, this paper uses utest command developed by Lind and Mehlum (2010) for reference.), indicating that there is a U-shaped relationship between environmental regulation and green innovation, which verifies Hypothesis 1a. This conclusion is consistent with the research results of Yi et al. (2018) [39]. Due to the scarcity of green innovation resources, the funds invested by enterprises in green innovation are limited. When the intensity of environmental regulation is weak, enterprises prefer to achieve environmental compliance through “minor repair and compensation” such as purchasing environmental protection equipment and reducing terminal emissions. Therefore, enterprises have insufficient motivation to rely on green innovation to compensate for environmental costs, and enterprise green innovation is restrained. However, after the intensity of environmental regulation exceeds the critical value, the cost to enterprises to maintain environmental responsibility gradually increases, and “minor repair and minor compensation” cannot meet environmental compliance needs. At this time, enterprises must strengthen green innovation to maintain environmental compliance and reduce environmental costs. From a strategic point of view, the scarcity of resources forces enterprises to invest limited funds in high-quality green innovation to quickly improve productivity and profitability to deal with the increasing cost of environmental regulation. Therefore, after the critical value is exceeded, environmental regulation will promote high-quality green innovation. Therefore, high-quality green innovation in enterprises has been encouraged by increasing the environmental regulation intensity after it has exceeded the critical value.

The impact of environmental regulation on low-quality green innovation did not pass the U test. According to Table 4 (2), there is a U-shaped relationship between environmental regulation and high-quality green innovation, which is consistent with the overall sample and verifies Hypothesis 1b. Nevertheless, according to Table 4 (3), the multiple linear regression results show that the correlation coefficient between environmental regulation intensity and low-quality green innovation was −0.1205, which was significant at the 1% level. This indicates that environmental regulation has a significant negative impact on low-quality green innovation and verifies Hypothesis 1c. In the case of limited R&D input funds, when the intensity of environmental regulation exceeds its critical value, enterprises prefer to invest in high-quality green innovation to offset environmental costs, further limiting their investment in low-quality green innovation such as utility models, which reduces the output of low-quality green innovation. This illustrates that high-quality green innovation has a crowding-out effect on low-quality green innovation.

From the empirical results of control variables, the growth ability (GRO) of enterprises has a negative impact on green innovation, but the results are not significant. This may be because growth ability is measured by the operating income growth rate, but under the influence of environmental regulation policies, the operating income growth rate is also affected by other external factors (such as environmental emission costs), resulting in its less significant impact on green innovation.

#### 5.2.2. Analysis of the Impact of ESG on Corporate Green Innovation

To study the impact of ESG on corporate green innovation, this study separately verified the impact of ESG comprehensive performance, environmental responsibility, social responsibility, and corporate governance on corporate green innovation (GIA). The test results are shown in Table 4 (3)–(6).

The influence coefficients of ESG, ERP, SRP, and GP on green innovation were 0.2898, 0.1512, 0.1166, and 0.2477, respectively, which were all significant at the 1% level, indicating that the better the ESG performance of the enterprise, the more it can promote green innovation. This verifies Hypothesis 2 and fully shows that not only the single variables of environmental responsibility, social responsibility and corporate governance can promote green innovation, but also the comprehensive performance of ESG can stimulate green innovation. The possible reason is that enterprises with better ESG performance pay more attention to the demands of shareholders, employees, the public, and other stakeholders and to establishing a good image of a responsible enterprise to effectively alleviate the normative pressure of the government and financial institutions on enterprises. The goal is for enterprises to obtain tax and financial incentives from the government and financial institutions to engage in green innovation. The more willing these enterprises are to improve their production process and production efficiency by increasing green innovation, the higher will be the degree of green innovation achieved. This is consistent with the research conclusion of Li et al. [14] and Xiang et al. [15].

#### 5.2.3. Analysis of the Results of the Regulatory Effect of ESG

To explore the regulatory role of ESG in determining the impact of enterprise environmental regulation intensity on green innovation, this paper examines the interaction between the linear and quadratic terms of environmental regulation and ESG comprehensive performance, environmental responsibility, social responsibility, and corporate governance. The test results are shown in Table 5 (1)–(4). The coefficient of the interaction term between the square of environmental regulation intensity and comprehensive ESG performance (ERI^2^ × ESG) was −1.2391; the coefficient of the interaction term between the square of environmental regulation intensity and environmental responsibility performance (ERI^2^ × ERP) was −0.7238; the coefficient of the interaction term between the square of environmental regulation intensity and social responsibility performance (ERI^2^ × SRP) was −0.4346; and the coefficient of the interaction term between the square of environmental regulation intensity and corporate governance (ERI^2^ × GP) was −2.6096, all of which were significant at the 1% level. These results indicate that ESG performance weakens the impact of environmental regulation on green innovation, which means that environmental regulation intensity and ESG performance have a substitution effect; thus, Hypothesis 3 has been verified. This implies that when environmental regulation does not exceed the critical value, ESG performance will reduce the negative impact of environmental regulation on green innovation, but that when environmental regulation intensity exceeds the critical value, ESG performance will inhibit the promotional effect of environmental regulation on enterprise green innovation. A possible reason for this is that companies with better ESG performance pay more attention to the interests of shareholders, employees, the public, and other stakeholders and the company’s social image. Their input and output levels in green innovation are high. With the increasing intensity of environmental regulation, these enterprises are more vulnerable to environmental compliance problems. Compared with other enterprises, when facing the same intensity of environmental regulation, enterprises with better ESG performance can obtain the same benefits with a lower level of green innovation to make up for environmental costs, or in other words, ESG performance has a buffer effect on enterprise green innovation.

### 5.3. Heterogeneity Analysis

In order to verify the differences in the impact of environmental regulation on Green Innovation of enterprises of different types and regions, as well as the differences in the regulatory effect of ESG among different enterprises, so as to provide suggestions for the government to formulate differentiated environmental regulation policies, this paper makes a heterogeneity analysis from the perspective of property right nature, ERI intensity and pollution degree.

#### 5.3.1. Heterogeneity Analysis Based on the Nature of Property Rights

This paper further distinguishes state-owned and non-state-owned enterprises and verifies the impact of environmental regulation intensity on green innovation in enterprises with different property rights. As shown in Table 6 (1)–(2), the influence coefficient of the quadratic term of environmental regulation intensity (ERI^2^) on green innovation in non-state-owned enterprises was 0.2641, which was significant at the 5% level, and the influence coefficient of ERI was −0.3387, which was significant at the 1% level, and their relationship passed the U test, indicating that the impact of environmental regulation intensity on green innovation of non-state-owned enterprises reflects the effect of “offset before compensation”. Nevertheless, the impact of environmental regulation intensity on green innovation of state-owned enterprises did not pass the U test. The influence coefficient of the linear relationship between them is −0.2023, which is significant at the level of 1%, indicating that environmental regulation intensity inhibits the green innovation of state-owned enterprises. This may be due to the good performance of state-owned enterprises in fulfilling their environmental responsibilities, the current intensity of environmental regulation only increases the cost of environmental compliance of state-owned enterprises and occupies the investment in green innovation, which distorts the resource allocation effect of environmental regulation more than the green technological innovation effect and inhibits the technological innovation power of state-owned enterprises.

#### 5.3.2. Heterogeneity Analysis Based on the Intensity of Environmental Regulation

In this study, areas with higher environmental regulation intensity than average were recorded as strong environmental regulation areas, and those with intensity lower than average were recorded as low environmental regulation areas. The intent was to further verify the impact of regional environmental regulation at different environmental regulation intensities on enterprise green innovation. As shown in Table 6 (3)–(4), the influence coefficient of the quadratic term of environmental regulation (ERI^2^) on green innovation in strong ERI areas was 8.6614, and the influence coefficient of the primary term (ERI) was −15.5122, which were respectively significant at the 5% and 1% level. However, in weak ERI areas, the influence coefficient of the quadratic term of environmental regulation intensity (ERI^2^) on green innovation was 0.3086, and the influence coefficient of ERI on green innovation was −0.4330, which were respectively significant at the 5% and 1% level. Clearly, the inflection point of enterprises in the weak ERI group is lower, and the angle of ascent is steeper. In other words, compared with the strong ERI areas, enterprises in the weak ERI areas are more sensitive to changes in environmental regulation intensity. This may be because enterprises in weak ERI areas have been under less pressure for environmental compliance for a long time. With the strengthening of environmental supervision, these enterprises will also turn to green innovation earlier to reduce environmental emissions.

#### 5.3.3. Heterogeneity Analysis Based on Pollution Degree

To analyze the impact of environmental regulation intensity on green innovation in enterprises with different pollution levels, this study further divided enterprises into heavily and lightly polluting groups. As shown in Table 6 (5)–(6), the influence coefficient of the quadratic environmental regulation intensity term (ERI2) of the lightly polluting group on green innovation was 0.3123, and the coefficient of the ERI was −0.4686, which was both significant at the 1% level, and their relationship passed the U test. Nevertheless, In the heavily polluting group, the influence coefficients of the environmental regulation intensity terms on green innovation were not significant. Therefore, the impact of environmental regulation intensity on green innovation in lightly polluting enterprises shows a significant U-shaped relationship, but the impact on heavily polluting enterprises is not significant. This may be because heavily polluting enterprises are subject to stricter supervision. In addition to the intensity of environmental regulation, green innovation in heavily polluting enterprises is also affected by many factors such as green credits (Hu et al., 2021), and therefore, they are less sensitive to the intensity of environmental regulation.

#### 5.3.4. Heterogeneity Analysis of Moderating Effects

To analyze the difference between the regulatory effect of ESG on the intensity of environmental regulation and green innovation in different enterprises, this study further tested by group according to the nature of property rights, the degree of environmental regulation, and the degree of pollution. The results show that ESG comprehensive performance (ESG), ERP, and GP have significant negative regulatory effects on both state-owned and non-state-owned enterprises, regional enterprises with strong and weak environmental regulation, and heavily polluting and lightly polluting enterprises, which further verifies Hypotheses 3 (Due to space constraints, these results will not be presented. Those who are interested can request it from the author.).

Social responsibility performance (SRP) still played a negative regulatory role in the process by which environmental regulation intensity affects green innovation, but there were obvious variations in different groups. The details are described below.

Table 7 (1)–(2) shows the test results for the regulatory effect when distinguishing the nature of property rights. The influence coefficient of the interactive term of squared environmental regulation intensity and social responsibility performance (ERI^2^ × SRP) on green innovation of non-state-owned enterprises was −0.4341 and was significant at the 10% level, and passed the U test. The results show that the better the performance of SRP, the smoother the U-shaped impact of environmental regulation intensity on green innovation. This implies that when environmental regulation does not exceed the critical value, SRP will reduce the negative impact of environmental regulation on green innovation of non-state-owned enterprises, but that when environmental regulation intensity exceeds the critical value, SRP will inhibit the promotional effect of environmental regulation on green innovation of non-state-owned enterprises. The environmental regulation intensity has a negative regulatory effect on the green innovation of state-owned enterprises, meaning that SRP weakens the negative impact of environmental regulation on the green innovation of state-owned enterprises.

Table 7 (3)–(4) shows the test results for the regulatory effect when distinguishing the intensity of environmental regulation. In weak ERI areas, the influence coefficient of the interactive term of the square of environmental regulation intensity and social responsibility performance (ERI^2^ × SRP) was −0.7702, which was significant at the 10% level, whereas in areas with strong ERI, the coefficient of the interaction term (ERI^2^ × SRP) was −0.5317 and did not pass the significance test. This shows that in weak ERI areas, the regulatory effect of social responsibility between environmental regulation intensity and green innovation is significant, but in strong ERI areas, this regulatory effect is not significant. In other words, the regulatory effect of SRP in weak ERI areas is better than in strong ERI areas. This may be because in areas with strong ERI, the impact on social responsibility performance is weak due to high environmental regulatory requirements. Therefore, when the intensity of environmental regulation changes, enterprises have to improve green innovation performance to offset the high environmental costs.

Table 7 (5) shows the test results for the regulatory effect of SRP on light pollution enterprises. The coefficient of the interactive term between the square of environmental regulation intensity and social responsibility performance in lightly polluting enterprises (ERI^2^ × SRP) was −0.4946, which was significant at the 1% level. This shows that social responsibility performance has a negative regulatory effect on light pollution enterprises, which is consistent with the previous conclusion.

## 6. Robustness Test

### 6.1. Endogenous Test: Instrumental Variable Method

In order to solve the endogenous problem that may lead to the bias of empirical results, this paper uses the instrumental variable method to re-estimate the 2SLS regression of the model. Many studies have shown that it is reasonable to use natural factors such as air circulation coefficient and atmospheric precipitation as instrumental variables [49]. In this paper, the logarithm of the average annual atmospheric precipitation in each region is used as the instrumental variable (the data comes from the CSMAR database, and the missing data is supplemented by interpolation and extrapolation). Because the atmospheric precipitation is only affected by regional climate conditions and has no correlation with other variables affecting green innovation, it meets the exclusive requirements.

The 2SLS regression estimation results are listed in Table 8. From the regression results of the first stage, there is a significant correlation between atmospheric precipitation and the intensity of environmental regulation. In the identification test of weak instrumental variables, the values of F statistics are greater than 10, and the estimation coefficients of atmospheric precipitation are significant at the 1% level, indicating that it is reasonable to select atmospheric precipitation as the instrumental variable. In the second stage, the 2SLS regression estimation results show that the “U” relationship between environmental regulation intensity and enterprise green innovation is still valid, and the results have passed the significance test.

### 6.2. Robustness Test of Empirical Results

In order to verify the robustness of the empirical results, this paper uses the data of the first lag period of green innovation as the alternative variable of green innovation, and uses the U test command to test whether the U-shaped relationship is established. The empirical results are shown in Table 9. It can be seen that the positive U-shaped relationship between the intensity of environmental regulation and high-quality green innovation is still established, and the intensity of environmental regulation has a negative impact on low-quality green innovation; ESG, ERP, SRP, and GP have a negative regulatory effect between environmental regulation and green innovation, which is consistent with the previous conclusion, indicating that the original hypothesis is robust.

## 7. Conclusions

Based on panel data from Chinese listed companies from 2010 to 2019, this paper explored whether the environmental regulation intensity and ESG performance have substitution effect on the impact of green innovation by constructing a double fixed effect model, and strived to provide new evidence for the further expansion and application of Porter Hypothesis. The main conclusions were as follows:(1)There is a U-shaped relationship between the intensity of environmental regulation and high-quality green innovation reflecting the effect of “offset before compensation”. While the low-quality green innovation is decreasing with the increasing intensity of environmental regulation, which shows that high-quality green innovation has a crowding out effect on low-quality.(2)It is found that not only the single variables of environmental responsibility, social responsibility, and corporate governance can promote green innovation, but also the comprehensive performance of ESG can stimulate green innovation.(3)The positive ESG performance shows a negative regulatory effect between environmental regulation intensity and enterprise green innovation, which means that environmental regulation intensity and ESG performance have a substitution effect.(4)There are significant differences in the impact of environmental regulation intensity on the green innovation of different enterprises. Firstly, the impact of environmental regulation intensity on green innovation of non-state-owned enterprises shows a U-shaped relationship, while environmental regulation intensity inhibits the green innovation of state-owned enterprises. Secondly, the intensity of environmental regulation has a stronger impact on green innovation in areas with weak environmental regulation. Thirdly, the impact of environmental regulation intensity on green innovation shows a U-shaped relationship in lightly polluting industries, but this relationship is not significant in heavily polluting industries.(5)From the perspective of regulatory effect, the regulatory effect of SRP is quite different. The better the performance of SRP, the smoother the U-shaped impact of environmental regulation intensity on green innovation of non-state-owned enterprises. SRP weakens the negative impact of environmental regulation on the green innovation of state-owned enterprises. The regulatory effect of SRP in weak ERI areas is better than in strong ERI areas, and SRP has a negative regulatory effect on lightly polluted enterprises.

## 8. Suggestions

Based on the research described above, this paper makes the following policy suggestions from the perspective of government and enterprises.

(1)From the perspective of the government

First, based on the differential impact of environmental regulation intensity on green innovation in different types of enterprises, the government should introduce more accurate environmental regulation policies to stimulate green innovation in enterprises of different types and in different regions by adjusting environmental regulation intensity.

Second, government regulators should constantly improve the ESG information disclosure system and should guide and encourage enterprises to conscientiously disclose environmental, social, and corporate governance information. On this basis, by improving the ESG information disclosure system, social capital can be gradually guided to flow into the field of green innovation, forming a virtuous circle to help enterprises develop green innovation and engage in sustainable development.

Third, the government should explore the introduction of an ESG reward and punishment system, including preferential policies such as tax reductions, subsidies, or discounts to those who achieve emission reduction through green technology innovation, and improved enterprise credit ratings to create the conditions for enterprises to carry out green innovation and achieve high-quality development. For example, the government can establish an ESG negative list system to punish enterprises with poor ESG performance through measures such as enterprise credit downgrades, reduction of loan amounts, and increases in bidding conditions.

(2)From the perspective of enterprises

First, enterprises should reshape their understanding of the ESG concept. By benchmarking the ESG concept, enterprises can focus their attention on increasing their level of high-quality green innovation and improving their own sustainable development abilities.

Second, awareness of ESG responsibilities by enterprise managers and employees should be improved. Enterprises need to build a green development culture, enhance awareness of enterprise management and employees to fulfill environmental and social responsibilities, strengthen corporate governance, and improve the level of green innovation.

## Figures and Tables

**Figure 1 ijerph-19-08558-f001:**
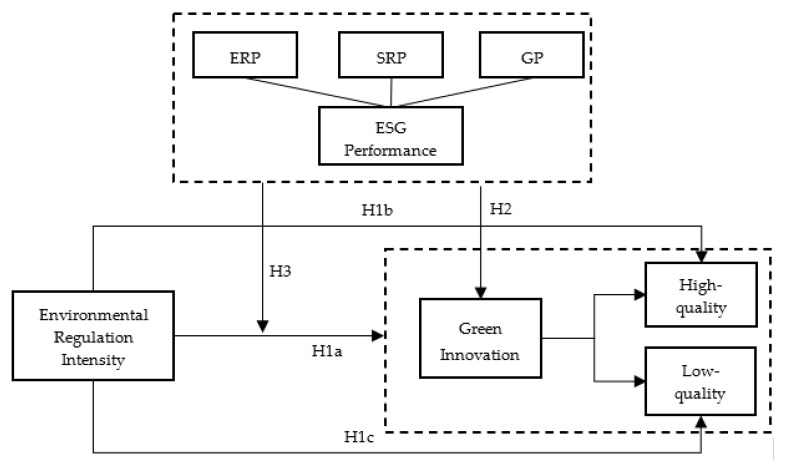
Theoretical model.

**Figure 2 ijerph-19-08558-f002:**
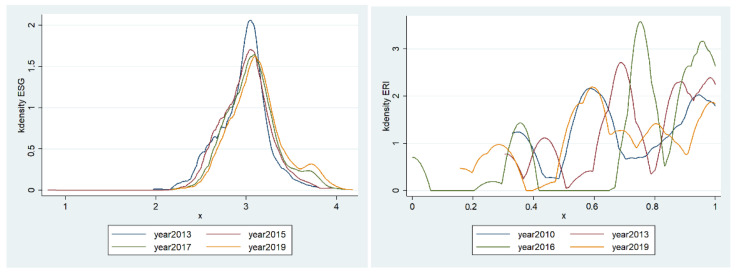
Kernel density chart of ESG performance and ERI.

**Table 1 ijerph-19-08558-t001:** Composition of ESG.

ESG	Environmental (33%)	Air Quality	4.78%
		Climate Change	4.70%
		Ecological & Biodiversity Impacts	4.79%
		Energy	4.73%
		Materials & Waste	4.74%
		Supply Chain	4.79%
		Water	4.79%
	Social responsibility (33%)	Community & Customers	5.53%
		Diversity	5.49%
		Ethics & Compliance	5.57%
		Health & Safety	5.58%
		Human Capital	5.55%
		Supply Chain	5.54%
	Governance (33%)	Audit Risk & Oversight	4.17%
		Board Composition	4.16%
		Compensation	4.16%
		Diversity	4.17%
		Independence	4.18%
		Nominations & Governance Oversight	4.18%
		Sustainability Governance	4.18%
		Tenure	4.18%

**Table 2 ijerph-19-08558-t002:** Definition of main variables.

Variable Types	Variable Name		Variable Symbol	Variable Definition	Unit of Measurement
**Dependent variables**	Green innovation		GIA	The logarithm of the number of green patents granted plus 1	Piece
	High-quality green innovation		HGIA	The logarithm of green invention patent authorization plus 1	Piece
	Low-quality green innovation		LGIA	The logarithm of green utility model patent authorization plus 1	Piece
**Independent variable**	Environmental Regulation Intensity		ERI	The comprehensive index of ERI calculated by the entropy method for the discharge of industrial wastewater, industrial sulfur dioxide, and industrial soot in each province (municipality, autonomous region)	%
**Moderator** **variables**	ESG performance	ESG Comprehensive Performance	ESG	ESG Composite Indices in Bloomberg Database	Score
		Environmental Responsibility Performance	ERP	Environmental Responsibility Index in Bloomberg Database	Score
		Social Responsibility Performance	SRP	Social Responsibility Index in Bloomberg Database	Score
		Governance Performance	GP	Governance Index in Bloomberg Database	Score
**Control** **variables**	Financial leverage		LEV	Asset-liability ratio, total liabilities/total assets	%
	Profitability		ROA	Net interest rate on total assets, net profit/total assets	%
	Growth ability		GRO	Operating income growth rate, operating income growth in the current period/total operating income at the beginning of the period	%
	Company value		TQ	Tobin’s Q, total market value at the end of the period/total assets at the end of the period	%
	Two duty unification		DUA	If the chairman and CEO are held by the same person, it is 1, otherwise, it is 0	1
	Proportion of independent directors		PID	Proportion of independent directors/directors	%
	Composite tax rate		CTR	(Business tax and surcharge + income tax expense)/total profit	%
	Property rights		PRO	Whether state-owned, 1 for state-owned enterprises, 0 for non-state-owned enterprises	1

**Table 3 ijerph-19-08558-t003:** Descriptive statistics.

Variables	Observation	Mean	S.D.	Min	Max
GIA	31,065	0.2984	0.7034	0.0000	6.7708
HGIA	31,065	0.1268	0.4469	0.0000	6.6026
LGIA	31,065	0.2273	0.6057	0.0000	5.7838
ERI	31,244	0.7197	0.2388	0.0000	1.0000
ESG	8990	3.0163	0.2982	0.8063	4.1762
ERP	7511	2.2869	0.5875	0.5739	4.1991
SRP	8742	3.1110	0.4039	1.5060	4.3592
GP	8990	3.8178	0.1175	1.5198	4.1827
LEV	27,270	0.4217	0.2109	0.0490	0.9016
GRO	26,070	0.4350	1.2266	−0.7396	9.1841
ROA	27,270	0.0387	0.0613	−0.2662	0.1934
TQ	26,279	1.0528	0.3540	0.5210	6.5750
DUA	26,940	0.2800	0.4490	0.0000	1.0000
CTR	31,346	0.3233	0.4021	−0.6418	2.5647
PID	27,222	0.3749	0.0557	0.1250	0.8000
PRO	27,270	0.3598	0.4799	0.0000	1.0000

**Table 4 ijerph-19-08558-t004:** Empirical results of the impact of ESG on corporate green innovation.

Variables	(1)	(2)	(3)	(4)	(5)	(6)	(7)
	Model 1	Model 1	Model 2	Model3			
	GIA	HGIA	LGIA	GIA	GIA	GIA	GIA
ERI	−0.394 *** (−3.84)	−0.4094 *** (−5.3997)	−0.1205 *** (−3.3606)				
ERI^2^	0.268 ** (2.93)	0.3642 *** (5.3867)					
ESG				0.2898 *** (10.0914)			
ERP					0.1512 *** (8.6814)		
SRP						0.1166 *** (5.2688)	
GP							0.2477 *** (3.6483)
LEV	0.124 *** (4.42)	0.0765 *** (3.6822)	0.0842 *** (3.1981)	−0.0025 (−0.0399)	−0.0628 (−0.8316)	−0.0113 (−0.1772)	0.0314 (0.5041)
GRO	−0.0002 (−0.07)	0.0012 (0.6163)	−0.0014 (−0.5546)	0.0013 (0.2089)	0.0041 (0.5818)	0.0028 (0.4545)	0.0012 (0.1892)
ROA	−0.230 *** (−3.85)	−0.2308 *** (−5.2229)	−0.0818 (−1.4614)	−0.3633 *** (−2.8984)	−0.5212 *** (−3.5370)	−0.3906 *** (−3.0387)	−0.4046 *** (−3.2108)
TQ	−0.0312 ** (−2.73)	−0.0142 * (−1.6907)	−0.0292 *** (−2.7498)	−0.0430 * (−1.6964)	−0.0357 (−1.1557)	−0.0497 * (−1.9172)	−0.0503 ** (−1.9699)
PRO	−0.0227 (−0.97)	−0.0301 * (−1.7447)	−0.0067 (−0.3082)	−0.1139 ** (−2.4405)	−0.1635 *** (−2.8623)	−0.1080 ** (−2.2823)	−0.0998 ** (−2.1269)
DUA	−0.0224 * (−2.30)	−0.0176 ** (−2.4478)	−0.0177 * (−1.9488)	−0.0159 (−0.8029)	−0.0030 (−0.1264)	−0.0208 (−1.0191)	−0.0199 (−0.9972)
CTR	0.0207 ** (2.76)	0.0197 *** (3.5526)	0.0117 * (1.6609)	0.0311 ** (2.1462)	0.0303 * (1.8641)	0.0285 * (1.9354)	0.0304 ** (2.0845)
PID	0.215 ** (2.72)	0.1212 ** (2.0801)	0.1291 * (1.7480)	0.3090 ** (2.1272)	0.3836 ** (2.3243)	0.3255 ** (2.1992)	0.3590 ** (2.4584)
α	0.346 * (2.54)	0.1253 (1.2466)	0.3052 ** (2.3967)	−0.4300 (−1.2894)	−0.1139 (−0.2309)	0.1059 (0.3189)	−0.5679 (−1.3523)
Entity fixed effects	Yes	Yes	Yes	Yes	Yes	Yes	Yes
Industry fixed effect	Yes	Yes	Yes	Yes	Yes	Yes	Yes
*N*	24839	24839	24839	8600	7202	8373	8600
r^2^	0.0062	0.0065	0.0039	0.0230	0.0231	0.0132	0.0114
F	2.59	2.7169	1.6527	4.1633	3.9061	2.2928	2.0350
U test(Slope-u)		0.3191 ***	−0.1091	-	-	-	-
U test(Slope-l)		−0.4094 ***	−0.1327	-	-	-	-
U test result		Yes	No	-	-	-	-

Robust t-statistics in parentheses; *** *p* < 0.01, ** *p* < 0.05, * *p* < 0.1. Slope-u indicates the upper bound of Slope value in the U teat result, and Slope-l indicates the lower bound of Slope value.

**Table 5 ijerph-19-08558-t005:** Empirical results of regulatory effect of ESG.

Variables	(1)	(2)	(3)	(4)
	Model 4			
ERI	−4.4670 ***(−7.2654)	−2.1471 ***(−6.2350)	−1.9039 ***(−3.6864)	−10.0529 ***(−5.0648)
ERI^2^	3.8188 ***(5.4939)	1.7697 ***(4.8631)	1.4233 **(2.4920)	10.0119 ***(4.6369)
ERI × ESG	1.3693 ***(6.9595)			
ERI^2^ × ESG	−1.2391 ***(−5.5480)			
ERI × ERP		0.7720 ***(6.5450)		
ERI^2^ × ERP		−0.7238 ***(−5.3762)		
ERI × SRP			0.4905 ***(3.1500)	
ERI^2^ × SRP			−0.4346 **(−2.4869)	
ERI × GP				2.5317 ***(4.8717)
ERI^2^ × GP				−2.6096 ***(−4.6170)
LEV	−0.0013(−0.0209)	−0.0541(−0.7119)	−0.0088(−0.1371)	0.0353(0.5624)
GRO	0.0020(0.3322)	0.0045(0.6304)	0.0037(0.5829)	0.0019(0.3003)
ROA	−0.3671 ***(−2.9120)	−0.5118 ***(−3.4570)	−0.3796 ***(−2.9373)	−0.3957 ***(−3.1262)
TQ	−0.0355(−1.3749)	−0.0269(−0.8549)	−0.0398(−1.5028)	−0.0366(−1.4085)
PRO	−0.1073 **(−2.2946)	−0.1587 ***(−2.7738)	−0.1040 **(−2.1924)	−0.0982 **(−2.0889)
DUA	−0.0149(−0.7472)	0.0003(0.0122)	−0.0206(−1.0027)	−0.0176(−0.8797)
CTR	0.0329 **(2.2614)	0.0313 *(1.9205)	0.0298 **(2.0151)	0.0328 **(2.2432)
PID	0.3195 **(2.1735)	0.3803 **(2.2796)	0.3298 **(2.2015)	0.3597 **(2.4371)
α	0.6785 **(2.0386)	0.4643(0.9313)	0.7508 **(2.2338)	0.7214 **(2.1578)
Entity fixed effects	Yes	Yes	Yes	Yes
Industry fixed effect	Yes	Yes	Yes	Yes
*N*	8544	7165	8319	8544
r^2^	0.0254	0.0256	0.0152	0.0161
F	4.2643	3.9983	2.4540	2.6779

Robust t-statistics in parentheses; *** *p* < 0.01, ** *p* < 0.05, * *p* < 0.1.

**Table 6 ijerph-19-08558-t006:** Heterogeneity test of the impact of ERI on green innovation.

Variables	(1)	(2)	(3)	(4)	(5)	(6)
	Model 1					
	State-Owned Enterprises	Non-State-Owned Enterprises	Strong ERI Group	Weak ERI Group	Heavily Polluting Group	Lightly Polluting Group
ERI	−0.2023 *** (−3.1524)	−0.3387 *** (−2.6380)	−15.5122 *** (−2.6102)	−0.4330 *** (−3.5063)	−0.0696 (−0.9000)	−0.4686 *** (−3.8275)
ERI^2^		0.2641 ** (2.2438)	8.6614** (2.5693)	0.3086** (2.4615)		0.3123 *** (2.8783)
LEV	−0.0209 (−0.4054)	0.1789 *** (5.1295)	0.1230 *** (3.0947)	0.1309 *** (3.1732)	−0.0084 (−0.1298)	0.1780 *** (5.5325)
GRO	0.0026 (0.6310)	−0.0017 (−0.4634)	0.0004 (0.1120)	0.0019 (0.4370)	0.0014 (0.1866)	−0.0005 (−0.1826)
ROA	−0.5143 *** (−4.4348)	−0.1446 ** (−2.0044)	−0.1939 ** (−2.2821)	−0.2715 *** (−3.0936)	−0.1752 (−1.2494)	−0.2739 *** (−4.0745)
TQ	−0.0493 ** (−2.2415)	−0.0168 (−1.2233)	0.0136 (0.8630)	−0.0624 *** (−3.6914)	−0.0388 (−1.3227)	−0.0300 ** (−2.4033)
PRO	−	−	−0.0005 (−0.0149)	−0.0458 (−1.2888)	0.0321 (0.5409)	−0.0404 (−1.5375)
DUA	−0.0080 (−0.4273)	−0.0253 ** (−2.1629)	−0.0168 (−1.2075)	−0.0262 * (−1.8785)	−0.0075 (−0.3298)	−0.0273 ** (−2.5037)
CTR	0.0311 *** (3.0691)	0.0143 (1.2776)	0.0293 *** (2.9589)	0.0166 (1.4254)	0.0235 (1.5668)	0.0209 ** (2.3738)
PID	0.4254 *** (3.6007)	0.0471 (0.4358)	0.2080 * (1.9523)	0.1909 (1.5945)	0.5575 *** (3.2551)	0.1265 (1.4005)
α	0.3532 ** (2.0645)	0.3024 (1.3370)	7.0465 *** (2.7198)	−0.0254 (−0.0645)	0.1046 (0.5754)	0.4466 *** (3.0651)
Entity fixed effects	Yes	Yes	Yes	Yes	Yes	Yes
Industry fixed effect	Yes	Yes	Yes	Yes	Yes	Yes
N	9202	15637	11136	13703	5799	19040
r^2^	0.0100	0.0070	0.0076	0.0086	0.0040	0.0086
F	2.2944	1.9230	1.8836	2.3533	0.9921	2.9679
U test(Slope-u)	0.07255	0.1894 *	1.8107 **	0.1842 *	0.1675	0.1560 *
U test(Slope-l)	−0.5325	−0.3387 ***	−15.51 ***	−0.4330 ***	−0.3153	−0.4686 ***
U test result	No	Yes	Yes	Yes	No	Yes

Robust t-statistics in parentheses; *** *p* < 0.01, ** *p* < 0.05, * *p* < 0.1. Slope-u indicates the upper bound of Slope value in the U teat result, and Slope-l indicates the lower bound of Slope value.

**Table 7 ijerph-19-08558-t007:** Heterogeneity test of the moderating effect of SRP.

Variables	(1)	(2)	(3)	(4)	(5)
	Model 4				
	State-Owned Enterprises	Non-State-Owned Enterprises	Strong ERI Group	Weak ERI Group	Lightly Polluting Group
ERI	−0.6258 *** (−4.0514)	−1.8223 ** (−2.5026)	−19.3973 * (−1.7574)	−2.4064 *** (−2.8469)	−2.1029 *** (−3.7100)
ERI^2^		1.3974 * (1.6794)	11.2652 * (1.7525)	2.3665 * (1.9275)	1.7155 *** (2.7578)
ERI × SRP	0.0829 ** (2.2600)	0.4940 ** (2.2333)	0.5943 (1.2563)	0.6776 ** (2.5625)	0.5125 *** (3.0319)
ERI^2^ × SRP		−0.4341 * (−1.6892)	−0.5317 (−1.0721)	−0.7702 ** (−1.9814)	−0.4946 *** (−2.6176)
LEV	−0.1060 (−1.1343)	0.0490 (0.5227)	0.0092 (0.1127)	−0.0624 (−0.5769)	0.1058 (1.4471)
GRO	0.0047 (0.5727)	0.0013 (0.1376)	0.0037 (0.4669)	0.0050 (0.4802)	0.0075 (1.1440)
ROA	−0.4729 ** (−2.3371)	−0.3013* (−1.7366)	−0.4313 *** (−2.5893)	−0.3431 (−1.6046)	−0.4758 *** (−3.2024)
TQ	0.0064 (0.1491)	−0.0565 * (−1.6496)	0.0133 (0.3842)	−0.0883 ** (−2.0702)	−0.0565 ** (−2.0184)
PRO	−	−	0.0100 (0.1665)	−0.2756 *** (−3.5028)	−0.1052 ** (−2.0732)
DUA	0.0274 (0.8681)	−0.0398 (−1.4154)	−0.0304 (−1.1196)	−0.0104 (−0.3213)	−0.0414 * (−1.8184)
CTR	0.0429 ** (2.4502)	0.0044 (0.1605)	0.0369 ** (2.0642)	0.0088 (0.3411)	0.0270 (1.5294)
PID	0.4849 *** (2.6736)	0.0340 (0.1261)	0.2526 (1.4381)	0.4055 (1.4619)	0.1583 (0.9365)
α	0.6844 * (1.9449)	0.5768 (0.7766)	8.2421 * (1.7305)	0.6615 *** (2.6168)	0.5748 (1.6363)
Entity fixed effects	Yes	Yes	Yes	Yes	Yes
Industry fixed effect	Yes	Yes	Yes	Yes	Yes
N	4366	3953	4260	4059	6136
r^2^	0.0156	0.0151	0.0178	0.0198	0.0212
F	1.8537	1.5386	1.9643	2.2730	2.9129

Robust t-statistics in parentheses; *** *p* < 0.01, ** *p* < 0.05, * *p* < 0.1.

**Table 8 ijerph-19-08558-t008:** Endogenous test: instrumental variables.

Variables	First-Stage		Second-Stage
	ERI	ERI^2^	GIA
AC	−0.4572678 *** (−24.13)	−0.5260457 *** (−24.35)	
AC^2^	0.0546091 *** (25.25)	0.0641621 *** (26.03)	
ERI			−2.22492 (−1.51)
ERI^2^			2.606446 ** (2.20)
Control variables	YES	YES	YES
Entity fixed effects	YES	YES	YES
Industry fixed effect	YES	YES	YES
F value	106.51	109.12	-
N	24756	24756	24756

Robust t-statistics in parentheses; *** *p* < 0.01, ** *p* < 0.05.

**Table 9 ijerph-19-08558-t009:** Robustness Test Results.

Variables	(1)	(2)	(3)	(4)	(5)	(6)
	Model 1	Model 2	Model3			
	HGIA-1	LGIA-1	GIA-1	GIA-1	GIA-1	GIA-1
ERI	−0.2858 *** (−3.5084)	−0.0173 (−0.4536)	−3.7111 *** (−5.5376)	−1.6780 *** (−4.4457)	−1.8095 *** (−3.2874)	−5.9173 *** (−2.6791)
ERI^2^	0.2628 *** (3.5603)		3.2065 *** (4.2285)	1.4103 *** (3.4842)	1.5028 ** (2.4606)	5.6748 ** (2.2653)
ERI × ESG			1.1731 *** (5.4187)			
ERI^2^ × ESG			−1.0532 *** (−4.2895)			
ERI × ERP				0.6404 *** (4.7054)		
ERI^2^ × ERP				−0.5943 *** (−3.8391)		
ERI × SRP					0.5207 *** (3.1027)	
ERI^2^ × SRP					−0.4709 ** (−2.5015)	
ERI × GP						1.4841 ** (2.5618)
ERI^2^ × GP						−1.4725 ** (−2.2404)
LEV	0.0693 *** (2.9175)	0.0926 *** (3.1174)	−0.0475 (−0.6978)	−0.1353 (−1.6133)	−0.0716 (−1.0216)	−0.0192 (−0.2799)
GRO	0.0007 (0.3361)	0.0006 (0.2194)	0.0056 (0.8490)	0.0071 (0.9344)	0.0067 (1.0062)	0.0057 (0.8568)
ROA	−0.1926 *** (−3.4658)	0.1416 ** (2.0373)	−0.0774 (−0.5103)	−0.2358 (−1.3233)	−0.1260 (−0.8094)	−0.1143 (−0.7510)
TQ	0.0087 (0.9391)	−0.0143 (−1.2324)	0.0012 (0.0423)	0.0142 (0.4216)	0.0031 (0.1123)	0.0010 (0.0349)
PRO	−0.0511 ** (−2.3285)	0.0329 (1.1978)	0.0446 (0.8221)	0.0369 (0.5472)	0.0406 (0.7388)	0.0516 (0.9466)
DUA	−0.0056 (−0.6893)	−0.0043 (−0.4190)	0.0127 (0.5745)	0.0391 (1.4403)	0.0145 (0.6379)	0.0100 (0.4483)
CTR	0.0117 * (1.8857)	−0.0036 (−0.4634)	0.0202 (1.2896)	0.0222 (1.2638)	0.0192 (1.2160)	0.0205 (1.3046)
PID	0.2038 *** (3.1017)	0.0622 (0.7568)	0.3395 ** (2.1099)	0.4502 ** (2.4489)	0.3635 ** (2.2264)	0.3658 ** (2.2643)
α	0.1101 (0.9728)	0.0346 (0.2445)	0.5104 (1.2752)	0.7130 (1.4183)	0.5356 (1.3405)	0.5368 (1.3360)
Entity fixed effects	Yes	Yes	Yes	Yes	Yes	Yes
Industry fixed effect	Yes	Yes	Yes	Yes	Yes	Yes
*N*	21209	21209	7384	6108	7165	7384
r^2^	0.0046	0.0034	0.0173	0.0177	0.0107	0.0091
F	1.7184	1.2997	2.7412	2.5247	1.6312	1.4265
U test(Slope-u)	0.2397 ***	0.0774	-	-	-	-
U test(Slope-l)	−0.2858 ***	−0.1167	-	-	-	-
U test result	Yes	No	-	-	-	-

Robust t-statistics in parentheses; *** *p* < 0.01, ** *p* < 0.05, * *p* < 0.1. Slope-u indicates the upper bound of Slope value in the U teat result, and Slope-l indicates the lower bound of Slope value.

## Data Availability

The data presented in this study are available on request from the Corresponding author. The data are not publicly available due to privacy.

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
