# Peer review of "Does the Environmental Regulation Intensity and ESG Performance Have a Substitution Effect on the Impact of Enterprise Green Innovation: Evidence from China"

_ijerph, 2022, doi:10.3390/ijerph19148558_

Round 1

Reviewer 1 Report

The paper addresses an important and timely issue regarding climate change. There are only a few minor comments that should be considered. These are as follows.

(1) One should more fully discuss in some detail the short- versus long-run effects.

(2) It would be helpful to explicitly say more about the benefits and costs.

(3) One should include the values for HGIA and LGIA in Table 3.

(4) One should consider including time fixed effects in Table 4. Also, did one cluster the standard errors? And the same comments apply to the other tables with empirical results.

(5) One should say something about whether the research design focuses on correlations or causation.

Reviewer 2 Report

This study examines ESG performance, environmental regulation intensity and enterprise green innovation. The research agenda is interesting and could contribute to the literature on environmental regulation intensity, ESG, green innovation.

The job is really well done; perhaps one could think for a future related work to also introduce a specific framework / theory in which to place the work.

Author Response

Dear Reviewer,

Thank you very much for your affirmation of the paper, which is the driving force for us to continue these studies. In addition, we have also revised the opinions of other reviewers, which can help us modify the paper more perfectly.

Thank you again for your hard work.

Best regards

Reviewer 3 Report

First, I would like to congratulate the authors for raising an interesting, despite overlooked, theme. The research is relevant, but it needs important re-structuring to grant the quality to be published.

The first obstacle I am concerned with is scientific soundness. The language style as well as the comment needs further refinement. Some sentences are very hard to follow and have too many ideas not linearly developed. 

The title needs re-structuring. 

Also, the abstract needs to provide the research question and the findings achieved, and the implications in both the theoretical and the practical terms. 

The introduction is long, but, still, does not link the tripod promised in the title. It is important to enlighten the reader about the connection among the topics raised. 

It seems to me that there is a need to grant the reader a literature review, or a theoretical debate. The reader needs to be aware about what was made so far and the "state of the art" in terms of the extant research. 

The underlying reason for including the variables in the equation needs to be justified. The present document presents the hypotheses without any contextualization. 

Also, the hypotheses need clarification. The empirical model does not test the hypotheses provided. There is a need to reformulate the hypotheses. The regulatory effect is an exogenous shock and is not affected by the firm performance (it is quite the other way around). 

The conceptual model is not coherent with the hypotheses proposed.  - need re-structurement. 

Variables need to be explored and justified. The variable table is presented and needs further comment. 

The equations should be re-formatted - please observe an econometrics book ( the multiplication signal should not be mentioned as well as the index in the random error). 

The purpose of the descriptive statistics is to add something about the database profile - this needs to be added to the main text.

Models need to be explained - there is no purpose, unless there is a strong reason, to maintain a variable in the equation that fails to be statistically different in all models (e.g. GRO). 

The results are not compared to previous literature, and there is plenty of evidence. 

Are the present results aligned with the previous findings? What is new?

The conclusions need to be strengthened. 

Please draw some implications for the theory and the practice. Are there some policy recommendations?

Good luck with your research!
